

# Dipole symmetry breaking and fractonic Nambu-Goldstone mode

Evangelos Afxonidis$^\star$, Alessio Caddeo$^\dagger$, Carlos Hoyos$^\ddagger$ and Daniele Musso$^\circ$

Department of Physics and Instituto de Ciencias y Tecnologías Espaciales de Asturias (ICTEA), Universidad de Oviedo, c/ Federico García Lorca 18, ES-33007 Oviedo, Spain

$\star$ evanafxon@gmail.com , $\dagger$ caddeoalessio@uniovi.es
$\ddagger$ hoyoscarlos@uniovi.es , $\circ$ mussodaniele@uniovi.es

## Abstract

We introduce a family of quantum field theories for fields carrying monopole and dipole charges. In contrast to previous realizations, fields have quadratic two-derivative kinetic terms. The dipole symmetry algebra is realized in a discretized internal space and connected to the physical space through a background gauge field. We study spontaneous symmetry breaking of dipole symmetry in 1+1 dimensions in a large-$N$ limit. The trivial classical vacuum is lifted by quantum corrections into a vacuum which breaks dipole symmetry while preserving monopole charge. By means of a Hubbard-Stratonovich transformation, heat-kernel and large-$N$ techniques, we compute the effective action for the low-energy modes. We encounter a fractonic immobile Nambu-Goldstone mode whose dispersion characteristics avoid Coleman-Hohenberg-Mermin-Wagner theorem independently of the large-$N$ limit.


# 1  Introduction

The spontaneous breaking of symmetries which depend on the coordinates is a topic which as yet lacks a complete and systematic understanding. In general, no universal rules relate the symmetry breaking pattern with the properties of the emerging low-energy Nambu-Goldstone modes, neither about their precise number, nor about their dispersion relations [1,2].

Multipole symmetries are a class of coordinate-dependent symmetries which combine an internal (monopole) symmetry with suitable polynomial combinations of the spatial coordinates. The monopole symmetry is an ordinary global symmetry whose conserved charge is a scalar with respect to rotations. When a higher moment of the charge is also conserved, like for instance dipole moment, quadrupole, *etc.*, we have a multipole symmetry. Multipole symmetries do not commute with translations since their charge densities depend explicitly on the coordinates. These symmetries impose restrictions on the mobility of isolated charged particles, thereby the theories enjoying multipole symmetries can have fractonic excitations. Fractons have sparked a lot of interest because of their possible application to quantum error correction in lattice models and extensions to field theories with unusual properties like an extensive degeneracy of states and IR/UV mixing, see [3–6] for reviews on the topic. It should be noted that they are not simply a theoretical curiosity, they can describe the low-energy properties of defects in quantum elasticity, see [7–15] for an incomplete list of references.

The spontaneous breaking of multipole symmetries has been recently investigated in [16–20]. Since the momentum density is not invariant under a multipole transformation, multipole symmetries are in general spontaneously broken in thermal states with a hydrodynamic description (including momentum conservation) [21–29].

Generally, it is assumed that the spontaneous breaking of the monopole/multipole symmetries produces a single scalar Nambu-Goldstone mode $\phi$, transforming as

$$\delta\phi = \lambda_0 + \lambda_{1i}x^i + \lambda_{2ij}x^i x^j + \cdots, \tag{1}$$

where $\lambda_0$, $\lambda_{1i}$, *etc.* are constant coefficients corresponding to the monopole, dipole and higher-moment transformations. It is interesting to appreciate that $\lambda_0$ and $\lambda_{1i}$ generate the same symmetry as that characterizing Galileons in flat space [30,31]. The most relevant low-energy terms in the Nambu-Goldstone effective action invariant under all multipole symmetries up to the $M$-th moment are

$$S_M = \int d^{d+1}x \left[ \frac{1}{2}(\partial_t\phi)^2 - \frac{\kappa^2}{2}(\partial^{M+1}\phi)^2 \right], \tag{2}$$

where, for even $M+1$, we have $\partial^{M+1} = (\nabla^2)^{\frac{M+1}{2}}$, while for odd $M+1$ we have $\partial^{M+1} = (\nabla^2)^{\frac{M}{2}}\partial_i$ and the spatial index is contracted such that the action is rotationally invariant. This is a Lifshitz theory with dynamical exponent $z = M + 1$, so that the Nambu-Goldstone mode

has a dispersion relation $\omega = \kappa |q|^{M+1}$. If the dimensionality of space is too low, then a multipole-symmetry generalization of the Coleman-Hohenberg-Mermin-Wagner (CHMW) theorem shows that spontaneous symmetry breaking is not possible in dimensions $d \leq 2z$ at non-zero temperature. However, dipole symmetry breaking at zero temperature is allowed even at $d = 1$ [32–35].

In a phase where not all the multipole symmetries up to order $M$ of the UV theory are spontaneously broken, the low-energy effective action is in general different from (2). For instance, for $M = 1$, when the dipole symmetry is spontaneously broken but the monopole symmetry is not, the order parameter is expected to have a spatial index and the Nambu-Goldstone mode is expected to exhibit a linear dispersion relation [18,33,36,37].

Focusing on continuous models with monopole and dipole symmetry, the concomitant spontaneous breaking of both symmetries follows in general the pattern described by Pretko [36]. There is a scalar order parameter charged under both monopole and dipole transformations,

$$\Phi \to \Phi' = e^{i(\lambda_0 + \lambda_{1i} x^i)} \Phi \,. \tag{3}$$

A dipole transformation is equivalent to a charge-dependent shift in momentum

$$\Phi'(x) = e^{i\lambda_{1i} x^i} \int \frac{d^d q}{(2\pi)^d} e^{iq_i x^i} \widetilde{\Phi}(q) = \int \frac{d^d q}{(2\pi)^d} e^{iq_i x^i} \widetilde{\Phi}(q - \lambda_1) = \int \frac{d^d q}{(2\pi)^d} e^{iq_i x^i} \widetilde{\Phi}'(q), \tag{4}$$

hence dipole symmetry is related to invariance under Galilean boost and UV/IR mixing.[1] The transformation properties of $\Phi$ are such that there cannot be an ordinary kinetic term compatible with dipole symmetry and the theory is "non-Gaussian" around $\Phi = 0$. In fact, the terms in the action have to depend on the difference of two momenta. On the other hand, when $\Phi$ acquires an expectation value, the effective theory for the Nambu-Goldstone mode falls in the class shown in (2), since both monopole and dipole symmetries are broken simultaneously. If one introduces a large number of fields transforming as in (3), it has been shown that at high temperature the dipole symmetry may be broken without breaking the monopole symmetry through the expectation value of a two-point function [37].

In [39] a different proposal for the realization of dipole symmetry was introduced, inspired by [40]. The dipole symmetry is introduced as an internal symmetry that implies a shift in momentum after a background field is turned on. Such background can be thought as emerging from a spontaneous symmetry breaking which locks the internal to the external space translations. Dipole symmetry is preserved by replacing ordinary derivatives by covariant derivatives and kinetic terms with the usual number of fields and derivatives are allowed. There is no reason a priori to expect that previous analyses of dipole symmetry breaking and the corresponding generalizations of CHMW theorem apply to this internal realization of dipole symmetry. The goal of the present paper is to build working models for the spontaneous breaking of dipole symmetry and examine in detail some of the questions mentioned above. In particular, we are interested in the case where monopole symmetry remains unbroken.

We will assume that there is an effective low-energy field theory description. It should be noted that lattice models with fractonic excitations may have different continuum limits, due to the IR/UV mixing. We will not be concerned by this type of questions and work directly in a continuum description without trying to address the UV origin of the model. All the scales present in the low-energy theory will be taken to be much smaller than the lattice cutoff.

The paper is organized as follows. Section 2 describes the realization of the monopole and dipole symmetries on charged scalar fields, with particular attention to the definition of suitable covariant derivatives. Section 3 constitutes the core of the paper. It first discusses a

---

[1]As discussed in [38], in this context the UV/IR mixing refers to the low-energy mixing among small and high momenta, something which characterizes Galilean hydrodynamics as well.

classical realization of the simultaneous breaking of monopole and dipole symmetries, whose effects comply with standard Nambu-Goldstone expectations. Then a quantum model where dipole symmetry is broken while preserving the monopole symmetry is thoroughly analyzed. We apply several effective-field-theory techniques involving auxiliary Hubbard-Stratonovich fields and the associated dualization, large-$N$ and saddle-point approximations and a heat-kernel expansion controlled by the scalar field mass, which sets the UV cut-off for the low-energy description. We also discuss in depth the properties of the emerging Nambu-Goldstone boson which is interestingly exotic: It is fractonic and completely immobile, a characteristic which makes it evade standard CHMW-theorem arguments. As a consequence, the symmetry breaking could occur also at zero temperature in $1 + 1$ dimensions, regardless of the large-$N$ suppression effects to the fluctuations of the order parameter. In section 4 we analyze a similar model for spinless fermions and in section 5 we conclude. Technical developments are detailed in the appendices.

## 2 Realization of dipole symmetry

Details about the algebra of generators of multipole symmetries in the context of fractons was originally discussed in [41]. In the simplest scenario, the relevant generators are those for spatial translations, $P_i$, the monopole charge $Q_0$, and the dipole charge $Q_1^i$. The only non-zero commutator is

$$i[P_i, Q_1^j] = \delta_i^j Q_0 \,. \tag{5}$$

This was dubbed the monopole-dipole-momentum algebra (MDMA) in [22], and it coincides with the centrally-extended Heisenberg algebra, as pointed out in [40].

The generators $P_i$ can be the usual translation operators shifting the spatial coordinates on which the fields depend

$$e^{i\boldsymbol{a}\cdot\boldsymbol{P}}\phi(t,\boldsymbol{x})e^{-i\boldsymbol{a}\cdot\boldsymbol{P}} = \phi(t,\boldsymbol{x}+\boldsymbol{a})\,. \tag{6}$$

As a consequence of the Stone-von Neumann theorem [42], all representations of the Heisenberg algebra are unitarily equivalent, thus the choice (6) does not reduce the generality of the present analysis. Completing to a unitary representation of the MDMA algebra, we have that the field $\phi(t,\boldsymbol{x})$ transforms as

$$e^{i\lambda_0 Q_0}\phi(t,\boldsymbol{x})e^{-i\lambda_0 Q_0} = e^{i\lambda_0}\phi(t,\boldsymbol{x})\,, \tag{7a}$$

$$e^{i\boldsymbol{\lambda}_1\cdot\boldsymbol{Q}_1}\phi(t,\boldsymbol{x})e^{-i\boldsymbol{\lambda}_1\cdot\boldsymbol{Q}_1} = e^{i\boldsymbol{\lambda}_1\cdot\boldsymbol{x}}\phi(t,\boldsymbol{x})\,. \tag{7b}$$

This leads to the type of action introduced by Pretko [36] where there is no quadratic term with space derivatives (when rotational symmetry is preserved).

Instead of using this realization which directly involves the space coordinates, we introduce a continuous set of complex fields $\phi_{\vec{X}}(t,\boldsymbol{x})$, labelled by a set of internal coordinates $X^I$, that are in a unitary Schrödinger representation of the Heisenberg group, namely

$$e^{i\vec{\kappa}\cdot\vec{P}}\phi_{\vec{X}}(t,\boldsymbol{x})e^{-i\vec{\kappa}\cdot\vec{P}} = \phi_{\vec{X}+\vec{\kappa}}(t,\boldsymbol{x})\,, \tag{8a}$$

$$e^{i\lambda_0 Q_0}\phi_{\vec{X}}(t,\boldsymbol{x})e^{-i\lambda_0 Q_0} = e^{i\lambda_0}\phi_{\vec{X}}(t,\boldsymbol{x})\,, \tag{8b}$$

$$e^{i\vec{\lambda}_1\cdot\vec{Q}_1}\phi_{\vec{X}}(t,\boldsymbol{x})e^{-i\vec{\lambda}_1\cdot\vec{Q}_1} = e^{i\vec{\lambda}_1\cdot\vec{X}}\phi_{\vec{X}}(t,\boldsymbol{x})\,. \tag{8c}$$

In principle, the space spanned by the internal coordinates $X^I$ can be of different dimensionality than the space spanned by the spatial coordinates $x^i$, but the usual dipole symmetry is recovered only when the internal and external spaces have the same dimensionality.

The symmetry (8) can be gauged by making the parameters of the transformations $\{\lambda_0, \vec{\lambda}_1, \vec{\kappa}\}$ dependent on the spacetime coordinates $(t, \boldsymbol{x})$. Besides, it is possible to define a covariant derivative

$$D_\mu \phi_{\vec{X}} = \partial_\mu \phi_{\vec{X}} - i a_\mu \phi_{\vec{X}} - i \vec{b}_\mu \cdot \vec{X} \phi_{\vec{X}} - \vec{V}_\mu \cdot \vec{\nabla}_X \phi_{\vec{X}} \,, \tag{9}$$

where, if $\hat{I}$ is the unit vector in the $I$-th direction, we have

$$\partial_{X^I} \phi_{\vec{X}} = \lim_{\epsilon \to 0} \frac{1}{\epsilon} \left( \phi_{\vec{X} + \epsilon \hat{I}} - \phi_{\vec{X}} \right) . \tag{10}$$

The gauge fields $\{a_\mu, \vec{b}_\mu, \vec{V}_\mu\}$ transform as [39]

$$\delta a_\mu = \partial_\mu \lambda_0 - \vec{V}_\mu \cdot \vec{\lambda}_1 + \vec{b}_\mu \cdot \vec{\kappa} \,, \tag{11a}$$

$$\delta \vec{b}_\mu = \partial_\mu \vec{\lambda}_1 \,, \tag{11b}$$

$$\delta \vec{V}_\mu = \partial_\mu \vec{\kappa} \,. \tag{11c}$$

With these gauge transformations, $D_\mu \phi_{\vec{X}}$ transforms in the same way as $\phi_{\vec{X}}$ in (8).

There can be fields in other representations of the MDMA algebra, for instance we can introduce fields $\chi_{\vec{X}}$ that carry no monopole charge and have a dipole charge $\vec{d}$:

$$e^{i\vec{\kappa} \cdot \vec{P}} \chi_{\vec{X}}(t, \boldsymbol{x}) e^{-i\vec{\kappa} \cdot \vec{P}} = \chi_{\vec{X} + \vec{\kappa}}(t, \boldsymbol{x}) \,, \tag{12a}$$

$$e^{i\lambda_0 Q_0} \chi_{\vec{X}}(t, \boldsymbol{x}) e^{-i\lambda_0 Q_0} = \chi_{\vec{X}}(t, \boldsymbol{x}) \,, \tag{12b}$$

$$e^{i\vec{\lambda}_1 \cdot \vec{Q}_1} \chi_{\vec{X}}(t, \boldsymbol{x}) e^{-i\vec{\lambda}_1 \cdot \vec{Q}_1} = e^{i\vec{\lambda}_1 \cdot \vec{d}} \chi_{\vec{X}}(t, \boldsymbol{x}) \,. \tag{12c}$$

Accordingly, the covariant derivative for $\chi_{\vec{X}}$ reads

$$D_\mu \chi_{\vec{X}} = \partial_\mu \chi_{\vec{X}} - i \vec{b}_\mu \cdot \vec{d} \, \chi_{\vec{X}} \,. \tag{13}$$

These fields are interesting because they allow us to write a different covariant derivative for $\phi_{\vec{X}}$, if one restricts to constant $\kappa$.[2] Consider a field $\chi_{\vec{X}}^I$ for each internal spatial dimension, namely $I \in \{1, ..., d\}$. If each field $\chi_{\vec{X}}^I$ carries a dipole charge $\vec{d}_I$ such that $(\vec{d}_I)^J = \delta_I^J$, then

$$D_\mu \phi_{\vec{X}} = \partial_\mu \phi_{\vec{X}} - i a_\mu \phi_{\vec{X}} - i \vec{b}_\mu \cdot \vec{X} \, \phi_{\vec{X}} - \phi_{\vec{X}} \sum_I V_\mu^I \, \log \chi_{\vec{X}}^I \,, \tag{14}$$

defines a covariant derivative for $\phi_{\vec{X}}(t, \boldsymbol{x})$, under the restriction $\partial_\mu \kappa = 0$. When the field $\chi_{\vec{X}}^I$ is not dimensionless there is a dimensionful factor inside the log that makes the argument dimensionless. For convenience we have set this factor to one.

Adopting either the covariant derivative (9) or (14), there is no obstruction to introducing quadratic derivative terms in the action with the internal realization of the symmetry that we have just discussed. The connection to the usual definition of coordinate-dependent dipole transformations emerges when the internal space spanned by the $X^I$ has the same dimensionality as the real space spanend by the $x^i$ and there is a constant background field

$$\vec{V}_0 = 0 \,, \qquad V_i^J = \delta_i^J V \,. \tag{15}$$

In this case there is a global symmetry transformation that leaves the gauge fields invariant

$$\lambda_0 = V \boldsymbol{a} \cdot \boldsymbol{x} \,, \qquad \lambda_{1I} = a_I \,, \tag{16}$$

with $\boldsymbol{a}$ a constant vector. Under this global transformation, the fields transform as

$$\phi_{\vec{X}} \to e^{iV\boldsymbol{a} \cdot \boldsymbol{x} + i\vec{a} \cdot \vec{X}} \phi_{\vec{X}} \,, \tag{17}$$

---

[2]This is actually a case considered below for a system with one internal, discretized spatial direction.

which amounts to a simultaneous shift of the phase linear both in the internal and external spatial coordinates.

In summary, the Schrödinger representation of the Heisenberg group in the internal space provides an alternative realization of dipole symmetry. However, according to (8) and (12), it involves a continuously-infinite number of fields because $X^I \in \mathbb{R}$, which makes unclear whether this realization can be treated as an ordinary field theory. Such issue can be avoided by generalizing the construction to a finite number $N$ of fields, obtained discretizing and compactifying the internal coordinates. In the remainder of the paper, we work in a large-$N$ limit, so we use the expressions for the transformations and covariant derivatives of the non-compact case, referring to a discrete though infinite number of fields. We discuss the compact case with a finite number of fields in appendix A.

We discretize the internal coordinates, as well as the associated translations,[3]

$$X^I \to n^I, \qquad \kappa^I \to k^I, \qquad n^I, k^I \in \mathbb{Z}, \tag{18}$$

hence the set of fields is infinite and numerable. The transformations (11) and (12) keep the same form also in the discretized case, they are obtained by simply replacing the continuous internal coordinates and translations with discrete integer-valued vectors $\vec{X} \to \vec{n}$, $\vec{\kappa} \to \vec{k}$. Importantly, since $\vec{k}$ is a vector of integers, $\delta \vec{V}_\mu = 0$. Moreover, the covariant derivative (9) involves derivatives with respect to $X^I$, which have to be replaced by a suitable discretized version. Using the unit vector in the $I$-th direction $\hat{I}$, we have

$$D_\mu \phi_{\vec{n}} = \partial_\mu \phi_{\vec{n}} - i a_\mu \phi_{\vec{n}} - i \vec{n} \cdot \vec{b}_\mu \phi_{\vec{n}} - \sum_I V_\mu^I \phi_{\vec{n}} \left[ \log(\phi_{\vec{n}}^* \phi_{\vec{n}+\hat{I}}) - \log(\phi_{\vec{n}}^* \phi_{\vec{n}}) \right]. \tag{19}$$

The continuous case is recovered by considering $\vec{X} = \epsilon \vec{n}$ and taking the limit $\epsilon \to 0$ such that[4]

$$\partial_{X^I} \phi_{\vec{X}} = \lim_{\epsilon \to 0} \frac{1}{\epsilon} \phi_{\vec{X}} \left[ \log(\phi_{\vec{X}}^* \phi_{\vec{X}+\epsilon \hat{I}}) - \log(\phi_{\vec{X}}^* \phi_{\vec{X}}) \right]. \tag{20}$$

The specific form of the logarithmic terms is chosen for later convenience. As we observed already in the continuous case, if there is a set of fields $\chi_{\vec{n}}^I$ which are not charged under monopole transformations but have dipole charge $(\vec{d}_I)^J = \delta_I^J$, then one can define a covariant derivative as

$$D_\mu \phi_{\vec{n}} = \partial_\mu \phi_{\vec{n}} - i a_\mu \phi_{\vec{n}} - i \vec{n} \cdot \vec{b}_\mu \phi_{\vec{n}} - \sum_I V_\mu^I \phi_{\vec{n}} \log(\chi_{\vec{n}}^I). \tag{21}$$

For simplicity, in the remainder of the paper we restrict to the case of a single spatial and internal direction.

## 3 Spontaneous breaking of the dipole symmetry

In the present section, we propose a model for bosonic complex scalar fields $\phi_n$ transforming in the one-dimensional discrete version of (8). First, we discuss the case in which the symmetry breaking is enforced by a classical potential and both monopole and dipole symmetries are broken simultaneously. Then, we consider the case in which the breaking is triggered by quantum fluctuations and only dipole symmetry is broken.

---

[3]Although the algebra is unchanged this does not necessarily correspond to a discretized version of the Heisenberg group.

[4]Reference [38] discusses the infinite volume limit at finite lattice spacing for theories with subsystem symmetries, a circumstance bearing a similarity to the discretization described here in the main text. Specifically, they connect the non-commutation of the infinite-volume and the continuum limits to the UV/IR mixing. Related comments, focused mainly on possible continuum descriptions (or lack thereof) for theories with subsystem symmetries, are given in [43].

### 3.1 Symmetry breaking with a classical potential

Let us consider a model with a classical Mexican-hat potential,

$$\mathcal{L} = \sum_{n=1}^{N} \left[ -|D_\mu \phi_n|^2 + m_\phi^2 |\phi_n|^2 - \frac{\lambda_\phi}{2} |\phi_n|^4 \right], \tag{22}$$

where we contract Lorentz indices with $\eta_{\mu\nu} = \mathrm{diag}(-1, 1)$ and

$$D_\mu \phi_n = \partial_\mu \phi_n - V_\mu \phi_n \left[ \log(\phi_{n+1} \phi_n^*) - \log(\phi_n \phi_n^*) \right]. \tag{23}$$

Actually, we are considering $a_\mu = b_\mu = 0$ and constant background field $V_\mu = (0, V_x)$. The model (22) displays monopole and dipole global symmetries. The former transformation corresponds to $(\lambda_0, \lambda_1) = (\alpha, 0)$ with constant $\alpha$, namely

$$\phi_n(t, x) \to e^{i\alpha} \phi_n(t, x). \tag{24}$$

The latter transformation corresponds to $(\lambda_0, \lambda_1) = (\beta x V_x, \beta)$ with $\beta$ constant,

$$\phi_n(t, x) \to e^{i\beta(n + x V_x)} \phi_n(t, x). \tag{25}$$

When the background $V_\mu$ is non-null, the dipole transformation (25) implies a shift in momentum, $\delta q = \beta V_x$. In fact, expanding $\phi_n$ in plane waves along the $x$ direction,

$$\phi_n(x) = \int \frac{dq}{2\pi} e^{iqx} \tilde{\phi}_n(q), \tag{26}$$

and performing a dipole transformation, we have

$$\phi_n'(x) = e^{i\beta(n + x V_x)} \int \frac{dq}{2\pi} e^{iqx} \tilde{\phi}_n(q) = e^{i\beta n} \int \frac{dq}{2\pi} e^{iqx} \tilde{\phi}_n(q - \beta V_x) = \int \frac{dq}{2\pi} e^{iqx} \tilde{\phi}_n'(q), \tag{27}$$

with

$$\tilde{\phi}_n'(q + \beta V_x) = e^{i\beta n} \tilde{\phi}_n(q). \tag{28}$$

We stress that the representation of dipole transformations described above is compatible with having an ordinary relativistic and quadratic two-derivative kinetic term , in contrast to the realization of [36]. The theory is nevertheless non-Gaussian due to the logarithmic terms appearing in the covariant derivative.

We are interested in a background for $V_\mu$ which breaks Lorentz boosts. Nevertheless, if there is spontaneous breaking of the monopole and dipole symmetries, we expect the Nambu-Goldstone boson to display a linear dispersion relation. Let us explicitly show this by considering configurations of the form

$$\phi_n(t, x) = \phi_{(0)n} e^{i\theta_n(t, x)}, \tag{29}$$

where $\phi_{(0)n}$ are the values of the fields at the classical vacuum and $\theta_n$ are their phase fluctuations. The potential obtained adding the non-derivative terms contained in $|D_\mu \phi_n|^2$ reads

$$\mathcal{V} = \sum_{n=1}^{N} \left[ V_x^2 \phi_{(0)n}^2 \left| \log\left( \frac{\phi_{(0)(n+1)}}{\phi_{(0)n}} \right) + i(\theta_{n+1} - \theta_n) \right|^2 - m_\phi^2 \phi_{(0)n}^2 + \frac{\lambda_\phi}{2} \phi_{(0)n}^4 \right]. \tag{30}$$

This potential is minimized when, for every $n$, we have $\phi_{(0)n} = \phi_{(0)} = \sqrt{\frac{m_\phi^2}{\lambda_\phi}}$, and $\theta_n = \theta$. The low-energy action for the phase amounts to

$$S_\theta = \frac{N m_\phi^2}{\lambda_\phi} \int d^2 x \left[ (\partial_t \theta)^2 - (\partial_x \theta)^2 \right]. \tag{31}$$

Therefore, the Nambu-Goldstone field $\theta$ displays a linear dispersion relation $\omega^2 = q^2$, differently to what happens in other realizations of the dipole symmetry.

Since the dipole symmetry (25) involves the spatial coordinate, the model displays degenerate coordinate-dependent vacuum configurations. The family of vacuum configurations is given by

$$\phi_n(t, x) = \phi_{(0)} e^{i\alpha} e^{i\beta(n + x V_x)} e^{i\theta(t,x)}, \tag{32}$$

where we have included the phase fluctuations $\theta(t, x)$. The constants $\alpha$ and $\beta$ parameterize a two-dimensional vacuum manifold. For $\beta \neq 0$ the configuration (32) does not minimize the potential $\mathcal{V}$ in (30), since $\theta_{n+1} - \theta_n = \beta \neq 0$, yet this is compensated by contributions from derivative terms. Taking these latter into account, the vacua (32) are all degenerate and the action for the Nambu-Goldstone field is given by (31), regardless of the specific vacuum configuration.

The Nambu-Goldstone field $\theta(t, x)$ realizes the global monopole and dipole symmetries non-linearly,

$$\theta(t, x) \quad \longrightarrow \quad \theta(t, x) + \alpha + \beta(n + x V_x). \tag{33}$$

The low-energy action (31) is invariant under these transformations only modulo a total spatial derivative term $2\beta \partial_x \theta$, which does not alter the equation of motion. According to (32), the transformations (33) are the rigid zero modes that connect different degenerate vacua. The Nambu-Goldstone particles corresponds to localized modulations of the Nambu-Goldstone field $\theta(t, x)$, for which we can drop the total derivative terms in the action.

The presence of just a Nambu-Goldstone mode despite the breaking of two symmetries can be expected a priori for two reasons. First, the rigid $n$-independent fluctuations about the vacuum are described by a complex field whose modulus and phase encode the fluctuations of the vacuum configuration $\phi_n(t, x)$. Since a constant in $x$ and $n$-independent variation of the modulus leads to a variation of the potential $\mathcal{V}$ while leaving the derivative terms unaffected, this represents a gapped Higgs mode. The remaining phase mode is necessarily gapless, according to the non-relativistic and spacetime generalizations of Goldstone theorem, which imply the presence of at least one Nambu-Goldstone mode when one or more symmetries are spontaneously broken [2, 44]. Secondly, the generators $\mathcal{Q}_0$ and $\mathcal{Q}_1$ for the global broken symmetries, respectively associated to $\alpha$ and $\beta$ in (33), satisfy the following commutation relation

$$[\Pi, \mathcal{Q}_1] = i\beta V_x \mathcal{Q}_0, \tag{34}$$

where $\Pi$ is the generator of the external spatial translations (*i.e.* the shifts in $x$). We recall that $V_x$ is a background field. Relying on the commutation relation (34), the present case is formally analogous to the presence of an inverse Higgs constraint [45].

In the present subsection, we enforced the spontaneous symmetry breaking of the dipole symmetry through the introduction of a classical potential. In the following, we discuss the case in which the spontaneous breaking of the dipole symmetry at zero temperature is triggered by quantum fluctuations.

## 3.2 Dipole symmetry breaking preserving monopole symmetry

We consider the action

$$\mathcal{L} = \sum_n \left( -|D_\mu \phi_n|^2 - m_\phi^2 \phi_n^* \phi_n \right) - \frac{m_\sigma^2}{2N} \left( \sum_n \phi_n^* \phi_n \right)^2$$

$$- \frac{\lambda_\sigma}{4N^3} \left( \sum_n \phi_n^* \phi_n \right)^4 - \frac{m_\chi^2}{N} \left| \sum_n \phi_{n+1} \phi_n^* \right|^2 - \frac{\lambda_\chi}{2N^3} \left| \sum_n \phi_{n+1} \phi_n^* \right|^4, \tag{35}$$

where $m_\phi^2 > 0$ while $m_\sigma^2$ and $m_\chi^2$ can have either sign. The action has a classical local minimum at $\phi_n = 0$, where all symmetries are unbroken. Below we show that, for some choices of the parameters, the large-$N$ quantum effective action associated to (35) can develop a stable saddle point where dipole symmetry is broken spontaneously while monopole symmetry is preserved.

### 3.2.1 Hubbard-Stratonovich transformation

We employ the Hubbard-Stratonovich transformation to express the action (35) into a more suitable form, where terms involving self-interactions of $\phi_n$ are replaced by terms involving Hubbard-Stratonovich auxiliary fields. We first identify non-quadratic terms in the action as combinations of the local (in $x$) composite operators $\sigma_n = \phi_n^* \phi_n$ and $\chi_n = \phi_{n+1}\phi_n^*$. We then promote these combinations into dynamical fields, introducing Lagrange multipliers $\tau_{\sigma_n}$ and $\tau_{\chi_n}$ to enforce the equivalence of the two partition functions,

$$
\begin{aligned}
Z &= \int \mathcal{D}\phi_n\, e^{iS[\phi_n]} \\
&= \int \mathcal{D}\phi_n \mathcal{D}\sigma_n \mathcal{D}\chi_n \mathcal{D}\tau_{\sigma_n} \mathcal{D}\tau_{\chi_n} \exp\Big( iS[\phi_n, \sigma_n, \chi_n] + i\int \sum_n \tau_{\sigma_n}(\sigma_n - \phi_n^* \phi_n) \\
&\quad + \tau_{\chi_n}^*(\chi_n - \phi_{n+1}\phi_n^*) + \tau_{\chi_n}(\chi_n^* - \phi_{n+1}^* \phi_n)\Big).
\end{aligned}
\tag{36}
$$

The transformed Lagrangian, including the Lagrange multipliers, reads

$$
\begin{aligned}
\mathcal{L}_{\text{HS}} &= \sum_n \Big( -|D_\mu \phi_n|^2 - m_\phi^2 \phi_n^* \phi_n \Big) - \frac{m_\sigma^2}{2N}\left(\sum_n \sigma_n\right)^2 \\
&\quad - \frac{\lambda_\sigma}{4N^3}\left(\sum_n \sigma_n\right)^4 - \frac{m_\chi^2}{N}\left|\sum_n \chi_n\right|^2 - \frac{\lambda_\chi}{2N^3}\left|\sum_n \chi_n\right|^4 \\
&\quad + \sum_n \Big[ \tau_{\sigma_n}\big(\sigma_n - \phi_n^* \phi_n\big) + \tau_{\chi_n}^*\big(\chi_n - \phi_{n+1}\phi_n^*\big) + \tau_{\chi_n}\big(\chi_n^* - \phi_{n+1}^* \phi_n\big)\Big].
\end{aligned}
\tag{37}
$$

Notice that the fields $\chi_n$ are charged under dipole symmetry while they are invariant under monopole transformations. The fields $\sigma_n$ are invariant under both symmetries.

The covariant derivative becomes

$$
D_\mu \phi_n = d_\mu \phi_n - V_\mu \phi_n \log\left(\frac{|\chi_n|}{\sigma_n}\right),
\tag{38}
$$

where, for later convenience, we introduced the notation

$$
d_\mu \equiv \partial_\mu - iV_\mu \theta_n,
\tag{39}
$$

with $\theta_n$ being the phase of $\chi_n$.

### 3.2.2 Effective action

We first assume that there are vacuum configurations that do not break internal translations, and introduce an $n$-independent ansatz for the auxiliary fields,

$$
\chi_n = \bar{\chi}, \qquad \sigma_n = \bar{\sigma}, \qquad \tau_{\chi_n} = \tau_\chi, \qquad \tau_{\sigma_n} = \tau_\sigma.
\tag{40}
$$

The Lagrangian simplifies to

$$\mathcal{L}_{HS} = \sum_n \left[ -|D_\mu \phi_n|^2 - \left( m_\phi^2 + \tau_\sigma \right) \phi_n^* \phi_n - \tau_\chi^* \phi_{n+1} \phi_n^* - \tau_\chi \phi_{n+1}^* \phi_n \right]$$
$$- N \left( \frac{m_\sigma^2}{2} \bar{\sigma}^2 + \frac{\lambda_\sigma}{4} \bar{\sigma}^4 + m_\chi^2 |\bar{\chi}|^2 + \frac{\lambda_\chi}{2} |\bar{\chi}|^4 - \tau_\chi^* \bar{\chi} - \tau_\chi \bar{\chi}^* - \tau_\sigma \bar{\sigma} \right). \tag{41}$$

Integrating by parts the kinetic term leads to

$$-|D_\mu \phi_n|^2 \longrightarrow \phi_n^* d_\mu d^\mu \phi_n - \xi \phi_n^* \phi_n, \tag{42}$$

where

$$\xi \equiv \tau_\sigma + \left( V_x \log \frac{|\bar{\chi}|}{\bar{\sigma}} \right)^2. \tag{43}$$

The fields $\phi_n$ enter the Lagrangian (41) quadratically. They can be thus integrated out to find a quantum effective action for the auxiliary fields. Crucially, the action for $\phi_n$ is such that the dependence in the phase of $\chi_n$ only enters as an effective gauge field in the 'covariant derivative' $d_\mu$ defined in (39). As we show explicitly below, this prevents terms $\sim |\partial_\mu \chi_n|^2$ from appearing in the low-energy effective action.

Defining

$$\Delta^2 = m_\phi^2 + \xi, \tag{44}$$

a standard calculation of the one-loop determinant (see appendix B) gives the low-energy effective Lagrangian for homogeneous (in $n$) configurations[5]

$$\mathcal{L}_{eff} = \frac{1}{48\pi m_\phi^4} \left[ (\partial_t \xi)^2 + 2 \partial_t \tau_\chi^* \partial_t \tau_\chi \right] - V_{eff}, \tag{45}$$

where

$$V_{eff} = \frac{m_\sigma^2}{2} \bar{\sigma}^2 + \frac{\lambda_\sigma}{4} \bar{\sigma}^4 + m_\chi^2 |\bar{\chi}|^2 + \frac{\lambda_\chi}{2} |\bar{\chi}|^4 - \tau_\chi^* \bar{\chi} - \tau_\chi \bar{\chi}^*$$
$$- \left[ \xi - \left( V_x \log \frac{|\bar{\chi}|}{\bar{\sigma}} \right)^2 \right] \bar{\sigma} - \frac{\Delta^2}{4\pi} \log \left( \frac{\Delta^2}{\Lambda^2} \right) + g(|\tau_\chi|^2, \Delta^2). \tag{46}$$

The symbol $\Lambda$ indicates a dynamically generated scale. For small values of $\frac{|\tau_\chi|^2}{\Delta^2}$, the last term can be expanded to quadratic order (see appendix B for details)

$$g(|\tau_\chi|^2, \Delta^2) = -\frac{|\tau_\chi|^2}{64\pi\Delta^2} + \mathcal{O}\left( \frac{|\tau_\chi|^4}{\Delta^4} \right). \tag{47}$$

### 3.2.3 Saddle point equations and symmetry-breaking solution

In the large-$N$ limit, the partition function is dominated by configurations that extremize the effective action

$$Z = \int \mathcal{D}\sigma_n \mathcal{D}\chi_n \mathcal{D}\xi_n \mathcal{D}\tau_{\chi_n} e^{iNS_{eff}[\sigma,\chi,\xi,\tau_\chi]} \sim e^{iNS_{eff}[\bar{\sigma}_0,\bar{\chi}_0,\xi_0,\tau_{\chi 0}]}, \tag{48}$$

---

[5] We define the effective action, so $\mathcal{L}_{eff}$ and $V_{eff}$ as well, extracting an $N$ factor, as we write in (48).

where, as already mentioned, we define the effective action factorizing a factor of $N$. The saddle point equations that determine these configurations are Euler-Lagrange equations obtained by varying (45) with respect to the auxiliary fields

$$\frac{1}{24\pi m_\phi^4}\partial_t^2 \xi_0 - \frac{1}{4\pi}\left[\log\left(\frac{e\Delta^2}{\Lambda^2}\right) - 4\pi\frac{\partial g}{\partial \Delta^2}\right] = \bar\sigma_0\,, \tag{49a}$$

$$\frac{1}{24\pi m_\phi^4}\partial_t^2 \tau_{\chi 0} + \frac{\partial g}{\partial \tau_{\chi 0}^*} = \bar\chi_0\,, \tag{49b}$$

$$-\xi_0 + V_x^2 \log\left(\frac{|\bar\chi_0|}{\bar\sigma_0}\right)\left[\log\left(\frac{|\bar\chi_0|}{\bar\sigma_0}\right) - 2\right] + m_\sigma^2 \bar\sigma_0 + \lambda_\sigma \bar\sigma_0^3 = 0\,, \tag{49c}$$

$$-\tau_{\chi 0} + V_x^2 \frac{\bar\sigma_0}{\bar\chi_0}\log\left(\frac{|\bar\chi_0|}{\bar\sigma_0}\right) + m_\chi^2 \bar\chi_0 + \lambda_\chi |\bar\chi_0|^2 \bar\chi_0 = 0\,. \tag{49d}$$

This system of equations admits time-independent solutions that break dipole symmetry ($|\chi| \neq 0$) for some range of values of the parameters in the effective potential. An analytic approximation to such solutions can be found if $|\tau_{\chi 0}|$ is small, so that $g(|\tau_{\chi 0}|^2, \Delta^2|)$ can be approximated by the simple form written in (47).

We introduce the following parameterization for $\chi$, $\sigma$ and $V_x$,[6]

$$\frac{|\bar\chi_0|}{\bar\sigma_0} = e^{-\beta+2}\,, \qquad \bar\sigma_0 = \frac{1-\alpha}{64\pi\beta}e^{2\beta-4}\,, \qquad V_x^2 = m_\phi^2(1-\alpha)v\,, \tag{50}$$

where $0 < \alpha < 1$. In order for the approximation (47) to be valid, we take $1-\alpha \ll 1$. Assuming that a solution exists, we solve the system (49) for $m_\chi^2$, $\Lambda^2$, $\xi_0$ and $\tau_{\chi 0}$, expanding in $1-\alpha$. Then we fix $m_\sigma^2$ so that the solution can be stable under time-dependent fluctuations (absence of tachyons). Stability corresponds to the values of $\beta$ and $v$ corresponding to the shaded region of Figure 1.

Following the procedure described above, we fix the parameters of the potential to

$$\frac{m_\sigma^2}{m_\phi^2} \simeq -4\pi\left[1 + \frac{32e^{4-2\beta}\beta\left(\beta^2-3\beta+2\right)v}{2\beta-3}\right] + \delta\mu_\sigma\,, \tag{51a}$$

$$\frac{m_\chi^2}{m_\phi^2} \simeq -64\pi\left[1 - v(\beta-2)\beta\right]$$

$$-\left\{\frac{4\pi\left[16e^4 v\beta(2\beta-1)(\beta-2)^2 + e^{2\beta}(3-2\beta)\right]}{e^4\beta(2\beta-3)} + \mathcal{O}(\delta\mu_\sigma)\right\}(1-\alpha)\,, \tag{51b}$$

$$\frac{\Lambda^2}{m_\phi^2} \simeq e + \left[\frac{ev(\beta-2)^2(2\beta-1)}{2\beta-3} + \mathcal{O}(\delta\mu_\sigma)\right](1-\alpha)\,, \tag{51c}$$

where $0 < \delta\mu_\sigma \ll 1$ is a small parameter. The solutions for the fields $\xi_0$ and $\tau_{\chi 0}$ up to order $(1-\alpha)$ are then given by

$$\frac{\xi_0}{m_\phi^2} \simeq -\frac{(4\pi-\delta\mu\sigma)e^{2\beta}(2\beta-3) - 64e^4\pi(\beta-2)^2\beta(2\beta-1)v}{64\pi e^4\beta(2\beta-3)}(1-\alpha)\,, \tag{52a}$$

$$\frac{\tau_{\chi 0}}{m_\phi^2} \simeq -\frac{1-\alpha}{\beta}e^{\beta-2}\,. \tag{52b}$$

Note that the solutions do not depend on the quartic couplings $\lambda_\sigma$ and $\lambda_\chi$, up to this order. However, without the quartic couplings the effective potential would not be bounded from

---

[6]Here, $\alpha$ and $\beta$ are different than the parameters in (32) and (33). No confusion should arise.

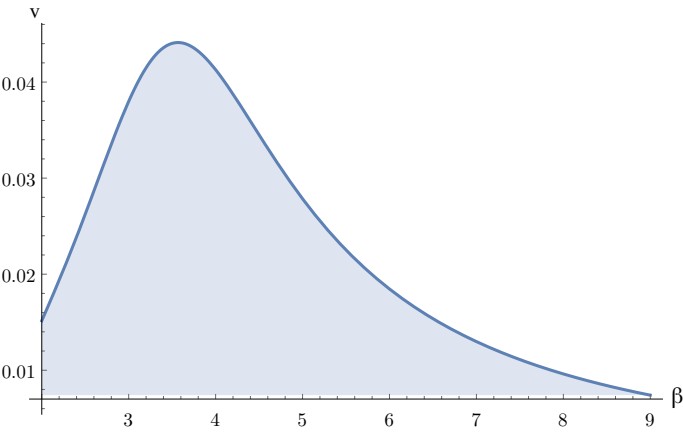

Figure 1: Values of $\beta$ and $v$ for which the dipole-symmetry-breaking solution is stable.

below, since $m_\sigma^2 < 0$ and $m_\chi^2 < 0$. Even introducing them, it is not guaranteed that the solution we have found is a global minimum. This is not an issue, since tunneling is suppressed by the large-$N$ limit. We can thus neglect non-perturbative instabilities.

Although we have traded four dimensionless parameters in the potential ($m_\sigma^2, m_\chi^2, \Lambda^2$ and $V_x^2$ in units of $m_\phi^2$) by other four parameters, ($\alpha, \beta, v$ and $\delta\mu_\sigma$), there is some degree of fine-tuning when we expand with respect to $V_x^2/m_\phi^2 \ll 1$, taking into account that $v$ is relatively small in the allowed range shown in Figure 1. Because of this, the values of the parameters for which solutions exist remains close to the values

$$m_\sigma^2 \simeq -4\pi m_\phi^2\,, \qquad m_\chi^2 \simeq -64\pi m_\phi^2\,, \qquad \Lambda \simeq e m_\phi^2\,. \tag{53}$$

### 3.2.4 Ground state and Nambu-Goldstone modes

We have found a stable large-$N$ saddle point of the quantum effective action which breaks spontaneously the dipole symmetry while preserving monopole symmetry. As we discussed, classical minima of the energy break both symmetries, with a single scalar Nambu-Goldstone mode with linear dispersion relation. Remarkably, when monopole symmetry is unbroken, we find that the Nambu-Goldstone mode does not have a linear dispersion relation. This key point is shown and discussed in detail below.

In the case at hand, the Nambu-Goldstone mode is the phase $\theta$ of the dipole-charged field $\bar{\chi}$. On general grounds, the low-energy theory describing the Nambu-Goldstone dynamics is symmetric with respect to shifts of $\theta$. The potential (46) is thereby independent of $\theta$ and the low-energy Lagrangian for $\theta$ involves derivatives terms only.

A one-loop calculation (see appendix B) gives the following Lagrangian density at the leading and next-to-leading orders in $\tilde{m}_\phi$

$$\mathcal{L}_{\text{NG}} = \frac{V_x^2}{240\,\pi\,\tilde{m}_\phi^4}\left[10\,\tilde{m}_\phi^2(\partial_t\theta)^2 - (\partial_x\partial_t\theta)^2 + (\partial_t^2\theta)^2\right]. \tag{54}$$

Here $\tilde{m}_\phi^2 = m_\phi^2 + \xi_0$, where $\xi_0$ is the background value of $\xi$. Note the $V_x^2$ in front of the action, one can understand its origin in that there is a global dipole symmetry only when $V_x \neq 0$.

The Lagrangian (54) does not involve terms with only spatial derivatives of the Nambu-Goldstone field. Indeed, every term involves at least two time derivatives. As we argue in appendix B below (B.21), this is actually true at any order in the heat-kernel expansion. Hence,

the low-energy action for the Nambu-Goldstone field $\theta$ features an emergent subsystem symmetry with respect to arbitrary spatial profiles $f(x)$, namely

$$\theta(t,x) \rightarrow \theta(t,x) + f(x). \tag{55}$$

From the low-energy perspective, the spatial profile of the Nambu-Goldstone field resembles a gauge redundancy. More specifically, the symmetry (55) prevents the Nambu-Goldstone to propagate, leading to immobile fractonic behavior.

The equation of motion up to this order are

$$10\tilde{m}_{\phi}^2 \, \partial_t^2 \theta + \partial_t^2 \partial_x^2 \theta - \partial_t^4 \theta = 0. \tag{56}$$

In principle, the corresponding dispersion relation presents two branches

$$\omega^2 = 0, \tag{57}$$
$$\omega^2 = q^2 - 10\tilde{m}_{\phi}^2, \tag{58}$$

but the branch (58) is beyond the low energy approximation $\omega^2, q^2 \ll m_{\phi}^2$ that we used to derive the effective action.

The low-energy effective Lagrangian (54) contains terms with higher time derivatives, such as $(\partial_t^2 \theta)^2$. Relying on the fact that they are sub-leading with respect to the dominant kinetic term $(\partial_t \theta)^2$, one can iteratively solve the equation of motion relegating the higher derivative terms to a neglected reminder, thus avoiding the issues related to Ostrogradsky instability [46, 47]. Note that this argument relies on the assumption of convergence of the heat-kernel expansion [48].

The branch (57) is an exactly flat band [33] whose flatness is due to the shift symmetry (55). It is fractonic in the sense that the propagation speed is exactly zero, so it corresponds to a mode that is immobile and can have an arbitrary spatial profile. In the context of effective theories of elasticity, this characteristics can be related to plasticity.

A similar situation arises from gradient-Mexican-hat models for the dynamical breaking of spatial translations, where the minimization of the potential implies (at least) an emergent subsystem symmetry and the associated Nambu-Goldstone mode is indeed a fractonic immobile phonon with $\omega^2 = 0$. When the emergent symmetry (55) is valid only at leading order in the derivative expansion, the Nambu-Goldstone dispersion $\omega^2 = 0$ can be deformed by higher-order corrections in the momenta which lead to propagation [49, 50].

### 3.2.5 Avoiding the Coleman-Hohenberg-Mermin-Wagner theorem

The CHMW theorem states that thermal (or quantum) fluctuations of the order parameter in three (two) or less spacetime dimensions are so large that they spoil the ordered vacuum. As a result, it is often claimed that there cannot be spontaneous symmetry breaking in these theories.

Let us recall one version of the argument. Assume that a symmetry is broken, in two dimensions and at zero temperature, by a scalar order parameter $\Phi$. In principle, there would be a Nambu-Goldstone boson $\theta$, which corresponds to fluctuations of the phase of the order parameter

$$\Phi = \langle|\Phi|\rangle \, e^{i\theta}. \tag{59}$$

Its low-energy effective action would be that of an ordinary massless field

$$S = \frac{f^2}{2} \int d^2x \left[ (\partial_t \theta)^2 - (\partial_x \theta)^2 \right], \tag{60}$$

where $f$ is a constant normalization factor. The strength of the quantum fluctuations of the order parameter is conveniently analyzed through the ratio

$$\rho(t,x) = \frac{\langle e^{i(\theta(t,x)-\theta(0,0))}\rangle}{\langle e^{i\theta(t,x)}\rangle\langle e^{-i\theta(0,0)}\rangle} = e^{\frac{1}{2}\langle\{\theta(t,x),\theta(0,0)\}\rangle}, \tag{61}$$

where we have used the fact that the action for the Nambu-Goldstone boson is Gaussian. If the symmetry is spontaneously broken, cluster decomposition in a local theory yields the factorization

$$\langle e^{i(\theta(0,x)-\theta(0,0))}\rangle \rightarrow \langle e^{i\theta(0,x)}\rangle\langle e^{-i\theta(0,0)}\rangle, \tag{62}$$

at large space separations [43]. Therefore, one expects the ratio defined in (61) to approach one

$$\lim_{|x|\to\infty}\rho(0,x) = 1. \tag{63}$$

On the other hand, if the correlations do not decay fast enough at long distances, $\rho(0,x)$ could be vanishing. It can be shown that for a massless field one gets

$$\rho(0,x) = (\mu|x|)^{-1/\pi f^2} \xrightarrow{|x|\to\infty} 0, \tag{64}$$

where $\mu$ is an arbitrary scale. The physical picture is that fluctuations are so strong that they destroy the ordered phase and there is no spontaneous symmetry breaking, namely $\langle\Phi\rangle = 0$.

The anti-commutator in the exponent of (61) equals the sum of the Wightman correlators $D^>(t,x) = \langle\theta(t,x)\theta(0,0)\rangle$ and $D^<(t,x) = \langle\theta(0,0)\theta(t,x)\rangle$ which, for a massless field, satisfy

$$(\partial_t^2 - \partial_x^2)D^{\lessgtr}(t,x) = 0. \tag{65}$$

The solutions are logarithmic functions that lead to the behaviour in (64).

Let us now study the effective low-energy theory for the Nambu-Goldstone modes described in subsection 3.2.4. We have the Lagrangian in (54) and the Wightman correlators for the Nambu-Goldstone field $\theta(t,x)$ satisfy

$$\left(10\tilde{m}_\phi^2\partial_t^2 + \partial_t^2\partial_x^2 - \partial_t^4\right)D^{\lessgtr}(t,x) = 0. \tag{66}$$

We work at low energy and momenta with a UV cut-off $\Lambda_0 < 10\tilde{m}_\phi^2$. In this regime, the first term in (66) dominates, reducing to the usual case (65), except for the absence of the spatial derivative term.

From the action (54), the retarded correlator is the solution to the equation

$$K\partial_t^2 D_R(t,x) = \delta(t)\delta(x), \qquad K = \frac{NV_x^2}{12\pi\tilde{m}_\phi^2}, \tag{67}$$

satisfying $D_R(t \leq 0, x) = 0$. The solution is

$$D_R(t,x) = \frac{t}{K}\Theta(t)\delta(x). \tag{68}$$

On the other hand, the retarded correlator is expressed in terms of the Wightman functions as

$$D_R(t,x) = i\Theta(t)\langle[\theta(t,x),\theta(0,0)]\rangle = i\Theta(t)\left[D^>(t,x) - D^<(t,x)\right]. \tag{69}$$

The Fourier transform in time then gives

$$\widetilde{D}_R(\omega,x) = \int_{-\infty}^{\infty} dt\, e^{i\omega t} D_R(t,x) = \frac{1}{\pi}\mathcal{P}\int_{-\infty}^{\infty} d\omega' \frac{\text{Im}\,\widetilde{D}_R(\omega',x)}{\omega'-\omega} + i\,\text{Im}\,\widetilde{D}_R(\omega,x), \tag{70}$$

where

$$\text{Im}\,\widetilde{D}_R(\omega, x) = \frac{1}{2}\left[\widetilde{D}^>(\omega, x) - \widetilde{D}^<(\omega, x)\right].\tag{71}$$

In (70) we recognize the Kramers-Kronig relation between the real and imaginary parts of the correlator. A direct calculation gives

$$\text{Im}\,\widetilde{D}_R(\omega, x) = -\frac{\pi}{K}\delta'(\omega)\delta(x).\tag{72}$$

Using now the properties $\widetilde{D}^>(\omega < 0, x) = 0$, $\widetilde{D}^<(\omega > 0, x) = 0$, and $D^<(-t, -x) = D^>(t, x)$, we obtain

$$D^>(t, x) = -\frac{it}{2K}\delta(x), \qquad D^<(t, x) = \frac{it}{2K}\delta(x).\tag{73}$$

Thus, the expectation value for the Nambu-Goldstone anti-commutator reads

$$\langle\{\theta(t, x), \theta(0, 0)\}\rangle = D^>(t, x) + D^<(t, x) = 0.\tag{74}$$

Therefore, from (61) and (74), we get

$$\lim_{|x|\to\infty}\rho(0, x) = 1.\tag{75}$$

As a result, our model admits spontaneous symmetry breaking at zero temperature, despite being defined in two spacetime dimensions.

## 4 Fermionic model with spontaneous breaking

In the present section, we propose another model that displays spontaneous dipole-symmetry breaking and no monopole-symmetry breaking at zero-temperature. The matter sector now involves fermionic (*i.e.* anticommuting) complex scalar fields $\psi_n$ transforming in the irreducible representation (8) and dipole fields $\chi_n$ transforming in the representation (12). Given its anticommuting character, $\psi_n$ satisfies

$$\psi_n\psi'_m = -\psi'_m\psi_n, \qquad (\psi_n\psi'_m)^* = (\psi'_m)^*\psi_n^*.\tag{76}$$

The covariant derivatives of these fields can be written (we consider the case $a_\mu = b_\mu = 0$) as

$$D_\mu\psi_n = \partial_\mu\psi_n - V_\mu\psi_n\log\chi_n,\tag{77a}$$
$$D_\mu\chi_n = \partial_\mu\chi_n - ib_\mu\chi_n.\tag{77b}$$

Let us consider the Lagrangian

$$\mathcal{L}_n = -|D_\mu\psi_n|^2 - m_\psi^2|\psi_n|^2 - |D_\mu\chi_n|^2 - m_\chi^2|\chi_n|^2,\tag{78}$$

where, similarly to what we have done in section 3, we assume that the mass parameters $m_{\psi_n}$ are the same for each $n$.

### 4.1 Effective potential

We integrate out the fields $\psi_n$ so to obtain an effective action for the dipole fields $\chi_n$. The computation is analogous to that of subsection 3.2 (more details can be found in appendix B). The fermionic Lagrangian after integration by parts is

$$\mathcal{L}_{\psi_n} = \psi_n^* d_\mu d^\mu\psi_n - (m_\psi^2 + \xi_n)\psi_n^*\psi_n,\tag{79}$$

where now

$$\xi_n = (V_x \log|\chi_n|)^2 + V_x \frac{\partial_x|\chi_n|}{|\chi_n|}. \tag{80}$$

The effective action for the dipole fields $\chi_n$ is defined through

$$
\begin{aligned}
e^{i\sum_n S_{\text{eff}}[\chi_n]} &= \int D\psi \, e^{i\sum_n \int d^2x \, \mathcal{L}_n[\psi_n,\chi_n]} \\
&= e^{i\sum_n \int d^2x \left(-|D_\mu\chi_n|^2 - m_\chi^2|\chi_n|^2\right)} \int D\psi \, e^{i\sum_n \int d^2x \, \psi_n^*\left(d_\mu d^\mu - m_\psi^2 - \xi_n\right)\psi_n} \\
&= e^{i\sum_n \int d^2x \left(-|D_\mu\chi_n|^2 - m_\chi^2|\chi_n|^2\right)} \prod_{n=1}^{N} \det\left(d_\mu d^\mu - m_\psi^2 - \xi_n\right),
\end{aligned}
\tag{81}
$$

where $D\psi$ indicates integration over all the fields $\psi_n$ and $\psi_n^*$. Notice that, since we are working with anticommuting fields $\psi_n$, the functional determinant appears with a positive power. Because of that, the resulting potential takes the same form of the second to last term in (46), but for a sign (see appendix B for details),

$$\Delta V_{\text{eff}\,n}(\chi_n) = \frac{\Delta_n^2}{4\pi}\log\left(\frac{\Delta_n^2}{\Lambda^2}\right), \qquad \Delta_n^2 = V_x^2(\log|\chi_n|)^2 + m_\psi^2. \tag{82}$$

The symbol $\Lambda$ represents a physical scale of the theory. Including as well the mass term, the effective potential reads

$$V_{\text{eff}\,n}(\chi_n) = \frac{1}{2}m_\chi^2|\chi_n|^2 + \frac{\Delta_n^2}{4\pi}\log\left(\frac{\Delta_n^2}{\Lambda^2}\right). \tag{83}$$

We have $N$ copies of the same effective action, so, for an $n$-independent configuration $\chi_n = \bar{\chi}$, the total effective potential is

$$N V_{\text{eff}}(\bar{\chi}) = \sum_{n=1}^{N} V_{\text{eff}\,n}(\bar{\chi}) = N\left[\frac{1}{2}m_\chi^2|\bar{\chi}|^2 + \frac{\Delta^2}{4\pi}\log\left(\frac{\Delta^2}{\Lambda^2}\right)\right]. \tag{84}$$

With the obvious notation $\Delta^2 = V_x^2(\log|\bar{\chi}|)^2 + m_\psi^2$.

The potential is depicted in Figure 2. For small values of $|\bar{\chi}|$, it diverges to positive infinity due to the logarithmic term. For large values of $|\bar{\chi}|$, instead, the mass term dominates, leading again to a positive divergence. Thereby, in the intermediate region, the potential features a global minimum. The model is hence guaranteed to exhibit spontaneous breaking of the dipole symmetry, triggered by the quantum fluctuations.

Since the dipole-symmetry-breaking configuration is a global minimum of the effective potential, for what concerns the stability of the vacuum, we do not need to assume the large-$N$ limit, in contrast to the bosonic case of section 3. However, as we show in subsection 4.2, the dispersion relation for the Nambu-Goldstone mode in this fermionic model is linear at small momenta as in the standard case. This is basically due to the fact that $\chi_n$ are dynamical fields and so the model Lagrangian includes a bare kinetic term for them. As a result, the CHMW theorem applies and there would not be spontaneous symmetry breaking in $1+1$ dimensions. Here we need to assume the large-$N$ limit in order to suppress the fluctuations of the Nambu-Goldstone boson that would otherwise spoil the ordered vacuum configuration.

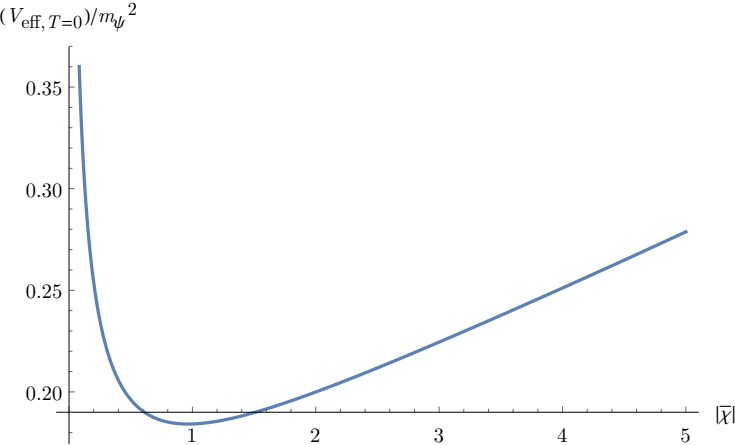

Figure 2: Effective potential for the fermionic model. Here we conveniently chose $(m_\chi/m_\psi)^2 = 0.002$, $(V_x/m_\psi)^2 = 0.1$, $(\Lambda/m_\psi)^2 = 0.1$.

## 4.2 Ground state and Goldstone modes

Having found that the fermionic model is characterized by an effective potential for the dipole field $\bar{\chi}$ that is minimized by a dipole-symmetry-breaking configuration $|\bar{\chi}| \neq 0$, at low energy the physics is captured by the small fluctuations $\theta$ around this vacuum configuration. The Nambu-Goldstone field $\theta$, defined as the phase of $\bar{\chi}$, enters the effective action only through derivative terms. Unlike the bosonic model of section 3, the dipole field $\bar{\chi}$ is dynamical at tree level. The Nambu-Goldstone action includes standard derivative terms as in (31). In addition to these, there are derivative terms coming from the one-loop determinant in (81) (see appendix B for more details). All in all, the Nambu-Goldstone Lagrangian reads

$$\mathcal{L}_{\text{NG}} = \left(1 - \frac{V_x^2}{24\pi\tilde{m}_\psi^2}\right)(\partial_t\theta)^2 - (\partial_x\theta)^2 - \frac{V_x^2}{240\pi\tilde{m}_\psi^4}\left[(\partial_t^2\theta)^2 - \partial_t^2\theta\partial_x^2\theta\right]. \tag{85}$$

Here $\tilde{m}_\phi^2 = m_\phi^2 + \xi_0$, where $\xi_0$ is the background value of $\xi$, defined in (80).

The equation of motion derived from this Lagrangian is

$$\left(1 - \frac{V_x^2}{24\pi\tilde{m}_\psi^2}\right)\partial_t^2\theta - \partial_x^2\theta + \frac{V_x^2}{240\pi\tilde{m}_\psi^4}\left(\partial_t^4\theta - \partial_t^2\partial_x^2\theta\right) = 0. \tag{86}$$

Going to Fourier space, the dispersion relation is

$$-\left(1 - \frac{V_x^2}{24\pi\tilde{m}_\psi^2}\right)\omega^2 + q^2 + \frac{V_x^2}{240\pi\tilde{m}_\psi^4}\left(\omega^4 - \omega^2 q^2\right) = 0, \tag{87}$$

which, being quadratic in $\omega^2$, features two branches. One is analogous to the higher branch encountered in (58); as such, it should not be considered in the low-energy theory. The dispersion relation of the relevant, lower branch is

$$\omega = c_1 q + c_3 q^3 + \mathcal{O}(q^5), \tag{88}$$

where

$$c_1 = \left(1 - v_x^2\right)^{-1/2}, \qquad c_3 = \frac{v_x^4\left(1 - v_x^2\right)^{-5/2}}{20\tilde{m}_\psi^2}, \qquad v_x^2 \equiv \frac{V_x^2}{24\pi\tilde{m}_\psi^2}. \tag{89}$$

Since in the present fermionic model the dipole field $\bar{\chi}$ is dynamical, the dispersion relation (88) of the Nambu-Goldstone field $\theta$ is not as exotic as that arising in the bosonic model (57). Indeed, at small momenta, it looks linear, with the speed of sound lowered by a quantum correction. As a result, the CHMW theorem holds in this case. To wit, in the fermionic model, the quantum fluctuations would spoil the ordered vacuum. As already mentioned, this can be avoided by taking the large-$N$ limit which suppresses the quantum fluctuations.

## 5 Conclusions

The present analysis introduces novel families of field theories, for either bosons or fermions, that are symmetric under dipole transformations and have ordinary (quadratic and with two derivatives) kinetic terms. We studied thoroughly some concrete examples that feature a spontaneous breaking of the dipole symmetry, with or without a concomitant breaking of monopole symmetry. The dispersion relations for the emerging Nambu-Goldstone modes differ from those encountered in other models studied before in the literature. Remarkably, we find a fractonic Nambu-Goldstone mode characterized by full immobility in a quantum model which preserves monopole symmetry. Such fractonic character of the low-energy mode evades the CHMW theorem in two spacetime dimensions at zero temperature. Therefore, the low-energy properties of systems with multipole symmetries are sensitive to the details of the symmetry realization and there might not be a universal effective description. Beyond the specific interest regarding multipole symmetry breaking, we provided a working counter-example to Coleman expectation. The exotic fractonic character of the Nambu-Goldstone mode being the crucial ingredient to escaping CHMW theorem. It would be interesting to explore the relationship of our result with possibly similar conclusions found in the context of non-Fermi liquids, where the avoidance of CHMW theorem is argued on the basis of a vanishing spectral weight for the gapless Nambu-Goldstone mode in the limit of small momenta [51].

The models analyzed in the present paper can be coupled to dynamical gauge fields for the monopole and dipole symmetries.[7] Besides, it would be interesting to extend the present analysis to three spacetime dimensions. With these extensions, one could address the low-energy description of dynamic elastic defects within the particle-vortex dual formulation proposed in [39] pursuing thus on the program reviewed in [14, 15] and recently revived by the connection to fractons [7].

## Acknowledgments

We would like to thank Riccardo Argurio, Erica Bertolini, Matteo Carrega and Nicola Maggiore for discussions and feedback.

**Funding information** The work of A.C. is partially supported by Fondazione Angelo Della Riccia and by Ministerio de Ciencia e Innovación de España under the program Juan de la Cierva-formación. This work is partially supported by the AEI and the MCIU through the Spanish grant PID2021-123021NB-I00 and by FICYT through the Asturian grant SV-PA-21-AYUD/2021/52177.

---

[7]The gauge structure of fracton theories has recently received attention in itself, see for example [52, 53].

# A  Generalization to a finite number of fields

Besides discretizing the internal space directions, in order to further reduce the set of fields to a finite number $N$, a possibility is that the discretized internal coordinates take values in $\mathbb{Z}_N$. This is akin to making the internal directions compact. It should be noted that in more than one dimensions this is not unique, as we could discretize a compact space in different ways, analogous to having different crystalline configurations. The compactification of the internal spatial directions does not lead to any modification of the gauge transformations or in the form of the covariant derivatives, but we should enforce

$$n^I + k^I \to (n^I + k^I) \bmod N. \tag{A.1}$$

In one dimension, $\phi_{\vec{n}} \to \phi_n$, it is also possible to generalize the transformations to a finite number of fields keeping $n$ as an ordinary integer, rather than a $\mathbb{Z}_N$-valued variable, with $0 \le n \le N - 1$. For convenience in the presentation, we henceforth replace $n = N$ with $n = 0$.

The finite transformations of the fields are the following:

$$e^{ikP}\phi_n(t,x)e^{-ikP} = \phi_{n+k}(t,x), \tag{A.2a}$$

$$e^{i\lambda_0 Q_0}\phi_n(t,x)e^{-i\lambda_0 Q_0} = \exp\left(i\lambda_0 \cos\frac{2\pi n}{N}\right)\phi_n(t,x), \tag{A.2b}$$

$$e^{i\lambda_1 Q_1}\phi_n(t,x)e^{-i\lambda_1 Q_1} = \exp\left(i\lambda_1 \frac{N}{2\pi}\sin\frac{2\pi n}{N}\right)\phi_n(t,x), \tag{A.2c}$$

where the trigonometric expressions encode the periodicity of the compactified space. More specifically, in the regime $n \ll N$, we recover a constant phase for the monopole transformation and an $n$-linear phase for the dipole transformation. The gauge fields $V_\mu$ are invariant under all gauge transformations and $a_\mu$ and $b_\mu$ transform as usual under monopole and dipole gauge transformations

$$\delta a_\mu = \partial_\mu \lambda_0 - V_\mu \lambda_1, \tag{A.3a}$$

$$\delta b_\mu = \partial_\mu \lambda_1. \tag{A.3b}$$

However, their transformation under translations is modified into

$$a_\mu \to \cos\frac{2\pi k}{N}a_\mu + \frac{N}{2\pi}\sin\frac{2\pi k}{N}b_\mu, \tag{A.4a}$$

$$b_\mu \to \cos\frac{2\pi k}{N}b_\mu - \frac{N}{2\pi}\sin\frac{2\pi k}{N}a_\mu. \tag{A.4b}$$

The covariant derivative takes the form

$$\begin{aligned}
D_\mu \phi_n = {}& \partial_\mu \phi_n - i\cos\frac{2\pi n}{N}a_\mu \phi_n - i\frac{N}{2\pi}\sin\frac{2\pi n}{N}b_\mu \phi_n \\
& - V_\mu \phi_n \frac{1}{\frac{N}{\pi}\sin\frac{2\pi}{N}}\left[\cos\frac{2\pi}{N}\log\left(\phi^*_{N-n}\phi_n\right) - \log\left(\phi^*_{N-n+1}\phi_{n-1}\right)\right].
\end{aligned} \tag{A.5}$$

Taking $N \to \infty$ with $n, k$ finite, one recovers the expressions for the non-compact case. When doing this one should consider $\phi_{N-n} = \phi_{-n}$.

A possible alternative to the realization given in (A.3) is:

$$\delta a_\mu = \partial_\mu \lambda_0 + V_\mu \lambda_0\left(1 - \cos\frac{2\pi}{N}\right) - V_\mu \lambda_1\left(1 + \sin\frac{2\pi}{N}\right), \tag{A.6a}$$

$$\delta b_\mu = \partial_\mu \lambda_1 - V_\mu \lambda_0 \frac{2\pi}{N}\sin\frac{2\pi}{N} + V_\mu \lambda_1 \frac{2\pi}{N}\left(1 - \cos\frac{2\pi}{N}\right). \tag{A.6b}$$

Together with the covariant derivative

$$D_\mu \phi_n = \partial_\mu \phi_n - i \cos\frac{2\pi n}{N} a_\mu \phi_n - i \frac{N}{2\pi} \sin\frac{2\pi n}{N} b_\mu \phi_n - V_\mu \phi_n \left[ \log\left(\phi_n^* \phi_{n+1}\right) - \log\left(\phi_n^* \phi_n\right)\right].$$
(A.7)

It is not completely straightforward to generalize this to a larger number of dimensions. A possibility is to introduce additional monopole gauge fields, one for each direction $a_\mu^I$. The expressions for the transformations of the fields are

$$e^{i\vec{k}\cdot\vec{P}} \phi_{\vec{n}}(t,\boldsymbol{x}) e^{-i\vec{k}\cdot\vec{P}} = \phi_{\vec{n}+\vec{k}}(t,\boldsymbol{x}),$$
(A.8a)

$$e^{i\lambda_0 Q_0} \phi_n(t,\boldsymbol{x}) e^{-i\lambda_0 Q_0} = \exp\left( i\lambda_0 \frac{1}{N} \sum_I \cos\frac{2\pi n^I}{N} \right) \phi_{\vec{n}}(t,\boldsymbol{x}),$$
(A.8b)

$$e^{i\vec{\lambda}_1 \cdot \vec{Q}_1} \phi_{\vec{n}}(t,\boldsymbol{x}) e^{-i\vec{\lambda}_1 \cdot \vec{Q}_1} = \exp\left( i \sum_I \lambda_1^I \frac{N}{2\pi} \sin\frac{2\pi n^I}{N} \right) \phi_{\vec{n}}(t,\boldsymbol{x}).$$
(A.8c)

The transformations of the gauge fields under monopole, dipole and translations in the $I$-th direction are

$$\delta a_\mu^I = \partial_\mu \lambda_0 - \vec{V}_\mu \vec{\lambda}_1,$$
(A.9a)

$$\delta \vec{b}_\mu = \partial_\mu \vec{\lambda}_1,$$
(A.9b)

$$a_\mu^I \rightarrow \cos\frac{2\pi k^I}{N} a_\mu^I + \frac{2\pi}{N} \sin\frac{2\pi k^I}{N} b_\mu^I,$$
(A.9c)

$$b_\mu^I \rightarrow \cos\frac{2\pi k^I}{N} b_\mu^I - \frac{N}{2\pi} \sin\frac{2\pi k^I}{N} a_\mu^I,$$
(A.9d)

$$a_\mu^{J\neq I} \rightarrow a_\mu^{J\neq I},$$
(A.9e)

$$b_\mu^{J\neq I} \rightarrow b_\mu^{J\neq I}.$$
(A.9f)

And the covariant derivative for these transformations is

$$D_\mu \phi_{\vec{n}} = \partial_\mu \phi_{\vec{n}} - i \frac{1}{N} \sum_I \cos\frac{2\pi n^I}{N} a_\mu^I \phi_{\vec{n}} - i \sum_I \frac{N}{2\pi} \sin\frac{2\pi n^I}{N} b_\mu^I \phi_{\vec{n}}$$
(A.10)

$$- \sum_I V_\mu \phi_{\vec{n}} \frac{1}{\frac{N}{\pi}\sin\frac{2\pi}{N}} \left[ \cos\frac{2\pi}{N} \log\left(\phi_{\vec{n}+(N-2n^I)\hat{I}}^* \phi_{\vec{n}}\right) - \log\left(\phi_{\vec{n}+(N-2n^I+1)\hat{I}}^* \phi_{\vec{n}-\hat{I}}\right)\right].$$

When $N \rightarrow \infty$, these also become the same as for the non-compact case, provided one identifies the monopole field as

$$a_\mu = \frac{1}{N} \sum_I a_\mu^I.$$
(A.11)

# B  One-loop computations

In the present appendix, we provide some details on the computations of the effective actions for the bosonic and fermionic models discussed, respectively, in sections 3 and 4.

### B.1   Bosonic model

After introducing Hubbard-Stratonovich fields, our bosonic model is described by the Lagrangian (41), which we report here for convenience,

$$\mathcal{L}_{\text{HS}} = \sum_n \left[ -|D_\mu \phi_n|^2 - \left( m_\phi^2 + \tau_\sigma \right) \phi_n^* \phi_n - \tau_\chi^* \phi_{n+1} \phi_n^* - \tau_\chi \phi_{n+1}^* \phi_n \right]$$
$$- N \left( \frac{m_\sigma^2}{2} \bar{\sigma}^2 + \frac{\lambda_\sigma}{4} \bar{\sigma}^4 + m_\chi^2 |\bar{\chi}|^2 + \frac{\lambda_\chi}{2} |\bar{\chi}|^4 - \tau_\chi^* \bar{\chi} - \tau_\chi \bar{\chi}^* - \tau_\sigma \bar{\sigma} \right). \quad (\text{B.1})$$

The second line in (B.1) gives a classical contribution to the effective action for the Hubbard-Stratonovich fields and the Lagrange multipliers, while the quantum contribution comes from evaluating the determinant arising from the Gaussian integrals in $\phi_n$, which can be written as[8]

$$N \Delta S_{\text{eff}} = i \operatorname{tr} \log \left[ \left( d^\mu d_\mu - \Delta^2 \right) \mathbb{1}_N + B_N \right]. \quad (\text{B.2})$$

Here,

$$\Delta^2 \equiv m_\phi^2 + \xi \equiv m_\phi^2 + \tau_\sigma + \left( V_x \log \frac{|\bar{\chi}|}{\bar{\sigma}} \right)^2, \quad (\text{B.3})$$

the symbol $\mathbb{1}_N$ denotes the identity matrix in $N$ dimensions and $B_N$ denotes the following $N \times N$-matrix

$$B_N = \begin{pmatrix} 0 & \tau_\chi & 0 & 0 & \cdots \\ \tau_\chi^* & 0 & \tau_\chi & 0 & \ddots \\ 0 & \tau_\chi^* & 0 & \tau_\chi & \ddots \\ 0 & 0 & \tau_\chi^* & 0 & \ddots \\ \vdots & \ddots & \ddots & \ddots & \ddots \end{pmatrix}. \quad (\text{B.4})$$

To compute the effective potential, we take constant field configurations, while, in order compute the derivative terms of the effective action, we use the heat-kernel approach.

**Effective potential**

Let us consider constant field configurations. From (B.2) we get

$$N \int d^2x \, \Delta V_{\text{eff}} = -iN \operatorname{tr} \log \left[ \left( d^\mu d_\mu - \Delta^2 \right) \right] - i \operatorname{tr} \log \left[ \mathbb{1}_N + \frac{B_N}{d^\mu d_\mu - \Delta^2} \right]. \quad (\text{B.5})$$

We perform the computation in Euclidean signature compactifying the time direction on a circle with length $\beta = 1/T$. We then take the zero-temperature limit $T \to 0$. Since here we are dealing with bosonic fields, we impose periodic conditions. In Euclidean signature, (B.5) reads

$$N \int d^2x \, \Delta V_{\text{eff}} = N \operatorname{tr} \log \left[ \left( -d_i d_i + \Delta^2 \right) \right] + \operatorname{tr} \log \left[ \mathbb{1}_N + \frac{B_N}{-d_i d_i + \Delta^2} \right]. \quad (\text{B.6})$$

Let us first consider the first contribution, we have explicitly

$$\operatorname{tr} \log(-d_i d_i + \Delta^2) = T \int \frac{dq}{2\pi} \log \left[ \prod_{k=-\infty}^{\infty} \left[ (2\pi T k)^2 + \Omega_q^2 \right] \right], \quad (\text{B.7})$$

---

[8]We use the notation tr for the trace over spacetime and internal space, while Tr will indicate the trace over the internal space only. We also recall that $S_{\text{eff}}$, $\mathcal{L}_{\text{eff}}$ and $V_{\text{eff}}$ are defined extracting a factor of $N$.

where $\Omega_q^2 \equiv (q - V_x \theta)^2 + \Delta^2$. We subtract the contribution of a free field dividing each factor in the argument of the log by $(2\pi T)^2 + \omega_q^2$ with $\omega_q^2 = q^2 + m_\phi^2$,

$$\prod_{k=-\infty}^{\infty} \frac{(2\pi Tk)^2 + \Omega_q^2}{(2\pi Tk)^2 + \omega_q^2} = \frac{\sinh^2\left(\frac{\Omega_q}{2T}\right)}{\sinh^2\left(\frac{\omega_q}{2T}\right)}. \tag{B.8}$$

Dropping the contribution that is independent of $\chi$, one then finds (neglecting also another constant term)

$$T \int \frac{dq}{2\pi} \log\left[\sinh^2\left(\frac{\Omega_q}{2T}\right)\right] = \int \frac{dq}{2\pi}\left[\Omega_q + 2T\log\left(1 - e^{-\Omega_q/T}\right)\right]. \tag{B.9}$$

The first term inside the square bracket can be identified as the zero-temperature contribution. We can evaluate it explicitly according to

$$\Omega_q = \frac{\mu}{\Gamma(-1/2)} \lim_{\epsilon \to 0}\left[-\frac{2}{\sqrt{\epsilon}} + \int_0^\infty d\tau (\tau + \epsilon)^{-3/2} e^{-(\tau+\epsilon)\Omega_q^2/\mu^2}\right], \tag{B.10}$$

where $\mu$ is a renormalization scale. Then, dropping the $\frac{1}{\sqrt{\epsilon}}$ divergent term, the integral over $q$ gives

$$\int \frac{dq}{2\pi} \Omega_q = -\frac{\mu^2}{4\pi} \lim_{\epsilon \to 0} \int_0^\infty d\tau (\tau + \epsilon)^{-2} e^{-(\tau+\epsilon)\Delta^2/\mu^2}. \tag{B.11}$$

Pursuing on, the integral over $\tau$ gives

$$\int \frac{dq}{2\pi} \Omega_q \sim -\frac{\mu^2}{4\pi\epsilon} + \frac{\Delta^2}{4\pi}(-\log\epsilon + 1 - \gamma_E) - \frac{\Delta^2}{4\pi}\log\left(\frac{\Delta^2}{\mu^2}\right) + \mathcal{O}(\epsilon). \tag{B.12}$$

Eventually, plugging this into (B.9) and suitably choosing the dynamically-generated scale $\Lambda$, we find that the first contribution to the one-loop effective potential in (B.6) is

$$\Delta V_{\text{eff}\,1, T=0} = -\frac{\Delta^2}{4\pi}\log\left(\frac{\Delta^2}{\Lambda^2}\right). \tag{B.13}$$

Let us now consider the second contribution in (B.6), recalling that $B_N$ is given by (B.4). For small values of $|\tau_\chi|/\Delta^2$, using that $\text{Tr}\, B_N = 0$ and $\text{Tr}\, B_N^2 = 2N|\tau_\chi|^2$ at large $N$, we have

$$\text{tr}\log\left[\mathbb{1}_N + \frac{B_N}{-d_i d_i + \Delta^2}\right] \approx -N|\tau_\chi|^2 \,\text{tr}\, \frac{1}{(d_i d_i + \Delta^2)^2} \tag{B.14}$$

$$= -N|\tau_\chi|^2 T \sum_k \int \frac{dq}{2\pi} \frac{1}{[(2\pi Tk)^2 + q^2 + \Delta^2]^2}$$

$$= -N|\tau_\chi|^2 \int \frac{dq}{2\pi} \frac{\left[\frac{\sqrt{\Delta^2+q^2}}{T} + \sinh\left(\frac{\sqrt{\Delta^2+q^2}}{T}\right)\right]\text{csch}^2\left(\frac{\sqrt{\Delta^2+q^2}}{2T}\right)}{8(\Delta^2 + q^2)^{3/2}}$$

$$= -N|\tau_\chi|^2 \int \frac{dq}{2\pi}\left[\frac{1}{64}\frac{1}{(q^2 + \Delta^2)^{3/2}} + \mathcal{O}\left(e^{-\sqrt{q^2+\Delta^2}/T}\right)\right].$$

As a result, in the zero-temperature limit we find

$$\text{tr}\log\left[\mathbb{1}_N + \frac{B_N}{-d_i d_i + \Delta^2}\right] \approx -N\frac{|\tau_\chi|^2}{64\pi\Delta^2}. \tag{B.15}$$

**Derivative terms**

To find the derivative terms explicitly, we adopt the heat-kernel approach, which consists in an expansion in powers of $1/m_\phi$ (see for instance [48,54]). It relies upon the identity

$$N\Delta\mathcal{L}_{\text{eff}} = i\langle x|\log\left[\left(d^\mu d_\mu - \Delta^2\right)\mathbb{1}_N + B_N\right]|x\rangle = -i\int_0^\infty \frac{d\tau}{\tau}H(x,\tau),\tag{B.16}$$

where we have introduced the heat kernel

$$H(x,\tau) \equiv \langle x|\exp\left(-\tau\left[\left(d^\mu d_\mu - \Delta^2\right)\mathbb{1}_N + B_N\right]\right)|x\rangle.\tag{B.17}$$

Equation (B.16) holds after removing an irrelevant infinite constant.

The one-loop effective action exhibits derivative terms for the fields $\xi$ and $|\tau_\chi|$ and for the Nambu-Goldstone field $\theta$ defined as the phase of the dipole-symmetry field $\chi$. For the stability analysis it will be enough to consider only time dependence. Then, these derivative terms are given by the following terms in the heat-kernel expansion

$$
\begin{aligned}
H_{\text{kin}}(\tau,x) &= i\frac{e^{-m_\phi^2\tau}}{48\pi}\tau^2\,\text{Tr}\left[(\partial_t\xi\mathbb{1} + \partial_t B_N)^2\right]\\
&= i\frac{e^{-m_\phi^2\tau}}{48\pi}\tau^2\left[N(\partial_t\xi)^2 + 2N\partial_t\tau_\chi^*\partial_t\tau_\chi\right],
\end{aligned}\tag{B.18}
$$

where we remind the reader that the field $\xi$ has been defined in (B.3). We also used that $\text{Tr}\,B_N = 0$ and $\text{Tr}\,B_N^2 = 2N|\tau_\chi|^2$ at large $N$. As a result, the kinetic terms in the low-energy effective Lagrangian read

$$N\mathcal{L}_{\text{kin}} = \frac{1}{48\pi m_\phi^4}\left[N(\partial_t\xi)^2 + 2N\partial_t\tau_\chi^*\partial_t\tau_\chi\right].\tag{B.19}$$

Let us consider the derivative terms for the Nambu-Goldstone field $\theta$. Recalling the form of the covariant derivative

$$D_\mu\phi_n = d_\mu\phi_n - V_\mu\phi_n\log\left(\frac{|\chi_n|}{\sigma_n}\right),\qquad d_\mu \equiv \partial_\mu - iV_\mu\theta_n,\tag{B.20}$$

and that we are considering $\theta_n = \theta$ (see discussion below (30)), we understand that $\theta$ enters the low-energy effective action only through terms involving the covariant derivative $d_\mu$. Since we expand around a background configuration, $\bar{\chi} = \bar{\chi}_0 + \bar{\chi}'$, we have $\xi = \xi_0 + \tilde{\xi}$, where $\xi_0$ is given by (B.3) evaluated on the background configuration. We then define $\tilde{m}_\phi^2 = m_\phi^2 + \xi_0$.

Given that $[\tilde{\xi},\theta] = 0$, the heat-kernel expansion does not produce interaction terms between the fluctuation fields $\tilde{\xi}$ and $\theta$. Thus, the relevant terms in the low-energy action which involve the Nambu-Goldstone field $\theta$ read (see [48] for the derivation of the heat kernel expansion)

$$
\begin{aligned}
H_{\text{NG}}(x,\tau) = iN\tau\frac{e^{-\tilde{m}_\phi^2\tau}}{48\pi}&\Big\{[d_\mu,d_\nu][d^\mu,d^\nu] + \frac{\tau}{15}\Big(4[d_\rho,[d_\mu,d_\nu]][d^\rho,[d^\mu,d^\nu]]\\
&+ [d_\nu,[d_\mu,d^\nu]][d_\rho,[d^\mu,d^\rho]] + 6[d_\rho,[d^\rho,[d_\mu,d_\nu]]][d^\mu,d^\nu]\Big) + \mathcal{O}(\tau^2)\Big\}.
\end{aligned}\tag{B.21}
$$

The gauge invariance of the original action (B.1) is preserved at each individual order of the heat kernel expansion. This implies that the covariant derivative $d_\mu$ can only appear through commutators with itself in nested structures which at their core have a commutator $[d_\mu, d_\nu]$.

Jointly with the fact that the background $V_\mu$ is only spatial, $V_\mu \propto \delta^x_\mu$, the inner commutators always produce a term $\partial_t \theta$. Hence, each Nambu-Goldstone field $\theta$ is always acted upon by (at least) one time derivative.

Evaluating the first terms of the large $\tilde{m}^2_\phi$ expansion of the heat kernel (B.21) and using (B.16), we find the leading contributions to Lagrangian density for the Nambu-Goldstone field,

$$\mathcal{L}_{\text{NG}} = \frac{V^2_x}{240 \, \pi \, \tilde{m}^4_\phi} \left[ 10 \, \tilde{m}^2_\phi (\partial_t \theta)^2 - (\partial_x \partial_t \theta)^2 + (\partial^2_t \theta)^2 \right]. \tag{B.22}$$

## B.2 Fermionic model

The effective potential for the fermionic model is found from

$$\int d^2 x \, \Delta V_{\text{eff}} = i \, \text{tr} \log \left[ \left( d^\mu d_\mu - \Delta^2 \right) \right], \tag{B.23}$$

which, in Euclidean signature, becomes

$$\int d^2 x \, \Delta V_{\text{eff}} = -\text{tr} \log \left[ \left( -d_i d_i + \Delta^2 \right) \right]. \tag{B.24}$$

For fermionic fields, we need to impose antiperiodic boundary conditions along the time circle. We have

$$-\text{tr} \log(-d_i d_i + \Delta^2) = -T \int \frac{dq}{2\pi} \log \left[ \prod_{k=-\infty}^{\infty} \left[ (\pi T(2k+1))^2 + \Omega^2_q \right] \right], \tag{B.25}$$

where $\Omega^2_q = (q - V_x \theta)^2 + \Delta^2$. Evaluating the sum over Matsubara frequencies, we find

$$\prod_{k=-\infty}^{\infty} \frac{[\pi T(2k+1)]^2 + \Omega^2_q}{[\pi T(2k+1)]^2 + \omega^2_q} = \frac{\cosh^2 \left( \frac{\Omega_q}{2T} \right)}{\cosh^2 \left( \frac{\omega_q}{2T} \right)}. \tag{B.26}$$

Dropping the contribution that is independent of $\chi$, one then finds (dropping another constant as well)

$$\Delta V_{\text{eff}} = T \int \frac{dq}{2\pi} \log \left[ \cosh^2 \left( \frac{\Omega_q}{2T} \right) \right] = \int \frac{dq}{2\pi} \left[ \Omega_q + 2T \log \left( 1 + e^{-\Omega_q/T} \right) \right]. \tag{B.27}$$

As a result, in the zero-temperature limit, we eventually obtain

$$\Delta V_{\text{eff},T=0} = \frac{\Delta^2}{4\pi} \log \left( \frac{\Delta^2}{\Lambda^2} \right). \tag{B.28}$$

**Derivative terms**

The derivative terms in the fermionic model can be computed again using the heat-kernel approach. Because of the anticommuting nature of the fermionic fields, we now have

$$\Delta \mathcal{L}_{\text{eff}} = -i \langle x | \log \left( d^\mu d_\mu - m^2_\phi - \xi \right) | x \rangle = i \int_0^\infty \frac{d\tau}{\tau} H(x, \tau), \tag{B.29}$$

where the heat kernel reads

$$H(x, \tau) \equiv \langle x | \exp \left( -\tau \left[ d^\mu d_\mu - m^2_\phi - \xi \right] \right) | x \rangle. \tag{B.30}$$

The quantum corrections to the Nambu-Goldstone effective Lagrangian is still found by evaluating (B.21). Including also the classical contributions, it reads

$$\mathcal{L}_{\text{NG}} = \left(1 - \frac{V_x^2}{24\pi\tilde{m}_\psi^2}\right)(\partial_t\theta)^2 - (\partial_x\theta)^2 - \frac{V_x^2}{240\pi\tilde{m}_\psi^4}\left[(\partial_t^2\theta)^2 - \partial_t^2\theta\,\partial_x^2\theta\right], \qquad \text{(B.31)}$$

where $\tilde{m}_\phi^2 = m_\phi^2 + \xi_0$, with $\xi_0$ being the background value of $\xi$.

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
