# Peer review of "Dipole symmetry breaking and fractonic Nambu-Goldstone mode"

_SciPost Physics, doi:SciPost Phys. Core 6, 082 (2023)_

## Round 1 · Referee Report · Anonymous · 2023-7-21

Strengths

1. The papers provides a new approach to field theories of fractons

Weaknesses

1. Some of the mathematical aspects of the paper are very heuristic

Report

The manuscript is devoted to field theories of excitations with mobility restrictions. It is a research direction that is worth pursuing. However, there are several points in the manuscript that need to be justified or explained better:
1) In section 2 a covariant derivative is introduced that contains a logarithm of a field. This leads to a restriction of the field and a large negative value of this derivative for small fields. In addition, the field seems to be always dimensionless. Is this true? If yes what is the physical interpretation of such a field.
2) The large N expansion seems to be effectively an expansion in large number of dimensions. Is there any connection between these two?
3) My intuition would be the discretization of the infinite dimensional representation of the algebra breaks the unitarity. I have not found a derivation in the paper, which shows that unitarity is preserved.
4) The part containing fermionic actions seems very heuristic. The authors should provide some justification the fermions can be realized in fracton theories e.g., by showing that fermionic representations of the algebra exist and are consistent with the action proposal in the paper.
5) Fractons tend to exhibit the UV/IR mixing, which makes the perturbative studies subtle. The authors use heat kernel approach in their perturbative studies. I have not found any discussion why using heat kernels is compatible with fractonic theories.

  • validity: ok
  • significance: good
  • originality: good
  • clarity: low
  • formatting: good
  • grammar: good

Author:  Daniele Musso  on 2023-07-26  [id 3838]

(in reply to Report 1 on 2023-07-21)

We thank the referee for carefully reading our paper and their pertinent comments. We provide a reply to the points raised in the report below:

1) For (2.10) the logarithmic term would indeed be divergent when the field with dipole charge ($\chi$) vanishes. This would imply that the action constructed with such derivative would diverge and therefore, the physical ground state would be at a configuration where either the field with dipole charge is non-vanishing $|\chi|\neq 0$, or the field on which the derivative is acting on is vanishing $\phi=0$. One might encounter similar cases when logarithmic terms are produced in the effective potential by loop corrections. In this regard, it is a dynamical constraint rather than a constraint that has to be imposed {\em a priori} over the field.

For (2.15) and related formulas the term in the covariant derivative with the logarithmic factor vanishes in the limit when the field goes to zero, since it is of the form $\phi \log |\phi|$.

The fields in the logarithm do not have to be necessarily dimensionless, there can be an additional scale inside the logarithm that we have omitted, we will comment this point in a revised version.

2) The large-$N$ expansion is actually an expansion in a large number of sites of discretized ``internal dimensions'', but the number of internal dimensions remains fixed and equal to the number of spatial dimensions.

3) This is an interesting question that we have not explored in full detail. First, let us point out that fields can be in non-unitary representations of non-compact groups, with the most evident example being fields in a finite dimensional representation of the Lorentz group.

In this case the discrete version of the transformations given in (2.4) are manifestly unitary, but it might be true is that these do not correspond to a discretized version of the Heisenberg group any more, even though they have the same algebra. We will comment on this point in a revised version.

4) We are discussing fermions as anti-commuting fields (as it is common in the context of non-relativistic theories), without any consideration regarding spinor representations, which might be what the referee has in mind. In this regard, fermions can be just in the same representations as the scalar fields.

5) As discussed for instance in 2201.10589 and 2205.01132, IR/UV mixing in theories with dipole symmetry is manifested when the theory is regularized on a lattice and reflects on the ground state degeneracy and periodicity properties of Goldstones associated to the breaking of dipole symmetry. This implies that one can reach different continuum theories depending on the properties of the lattice. We work in a regime where the continuum limit has been already taken but there are additional scales (much smaller than the lattice scale) so that one can study the low energy effective theory below these additional scales using ordinary field theory techniques. We will add a paragraph explaining this point.

---

## Editorial Decision

published